# Correlations between Electrophysiological Parameters, Lymphocyte Distribution and Cytokine Levels in Patients with Chronic Demyelinating Inflammatory Polyneuropathy

**DOI:** 10.3390/jpm11080766

**Published:** 2021-08-04

**Authors:** Edyta Dziadkowiak, Helena Moreira, Malgorzata Wieczorek, Slawomir Budrewicz, Ewa Barg, Magdalena Koszewicz

**Affiliations:** 1Department of Neurology, Wroclaw Medical University, Borowska 213, 50-556 Wroclaw, Poland; edyta.dziadkowiak@umed.wroc.pl (E.D.); slawomir.budrewicz@umed.wroc.pl (S.B.); 2Department of Basic Medical Sciences, Wroclaw Medical University, Borowska 211, 50-556 Wroclaw, Poland; helena.moreira@umed.wroc.pl (H.M.); ewa.barg@umed.wroc.pl (E.B.); 3Faculty of Earth Sciences and Environmental Management, University of Wroclaw, Uniwersytecki 1, 50-137 Wroclaw, Poland; malgorzata.wieczorek@uwr.edu.pl

**Keywords:** typical CIDP, lymphocyte distribution, cytokine levels, axonal degeneration

## Abstract

The goal of this study was to analyse, in relation to electrophysiological results, the distribution of lymphocyte subpopulations and the level of cytokines in patients with the typical form of chronic demyelinating inflammatory polyneuropathy (CIDP) before immunoglobulin treatment. The study group consisted of 60 patients (52 men, eight women), with a mean age 64.8 ± 11.2, who fulfilled the diagnostic criteria for the typical variant of CIDP, with (23 patients) and without (37 patients) diabetes mellitus. We analysed the results of the neurophysiological tests, and correlated them with the leukocyte subpopulations, and cytokine levels. In CIDP patients, IL-6, IL-2, IL-4 and TNF-α levels were significantly increased compared to the control group. Fifty patients had decreased levels of T CD8+ lymphocytes, and 51 patients had increased levels of CD4+ lymphocytes. An increased CD4+/CD8+ ratio was also found. Negative correlations were observed mainly between compound muscle action potential (CMAP) amplitudes and cytokine levels. The study enabled the conclusion that electrophysiological parameters in CIDP patients are closely related to the autoimmune process, but without any clear differences between patients with and without diabetes mellitus. Correlations found in the study indicated that axonal degeneration might be independent of the demyelinating process and might be caused by direct inflammatory infiltration.

## 1. Introduction

Inflammatory neuropathies include the group of heterogeneous disorders characterized by immune-mediated hematogenous leukocyte infiltration of peripheral nerves, nerve roots or both, causing demyelination or axonal degeneration or both. Chronic acquired demyelinating neuropathies (CADP) are an important group of immune neuromuscular disorders affecting myelin [1]. Three dominant disease categories can be distinguished among inflammatory neuropathies: Guillain–Barré syndrome, chronic inflammatory demyelinating polyradiculoneuropathy (CIDP) and non-systemic vasculitic neuropathy (or peripheral nerve vasculitis) [2,3]. Another rare immune-mediated neuropathy is multifocal motor neuropathy (MMN), characterized by slowly progressive, asymmetric, detached paresis of mainly the upper extremities. [1,4,5]. Additionally, CIDP may coexist with other systemic diseases such as diabetes mellitus (DM) and monoclonal gammopathy of uncertain significance (MGUS). Chronic inflammatory demyelinating polyneuropathy in diabetic patients is 11 times more common than in the general population. It is also suspected that DM predisposes to CIDP. CIDP associated with diabetes mellitus (CIDP-DM) differs from the CIDP subtype without diabetes in terms of clinical symptoms, laboratory results, response to therapy, and prognosis. CIDP-DM is a serious disease, but it responds significantly better to corticosteroids or intravenous immunoglobulin treatment and has fewer relapses than non-diabetic CIDP [6,7,8]. Despite major advances in molecular biology, pathology and genetics, the pathogenesis of these disorders remains ambiguous. Diet, consumption of alcohol, tobacco usage, and dysbiosis also seem to exert a strong influence on neuroinflammatory processes [9].

The abiding theory of CIDP pathogenesis is that cell-mediated and humoral mechanisms act synergistically to cause damage to peripheral nerves. Antibodies against peripheral nerve myelin proteins or node of Ranvier components are too infrequently detected in the sera of CIDP patients to be considered pathogenic or molecular markers of the disease. IgG or IgM antibodies against peripheral nerve myelin 22 (0–50%), myelin basic protein P2 (11–35%), myelin protein zero (0–30%), connexin 32 and the myelin basic protein are detected in about 2.5% of patients with CIDP, with frequencies no higher than in healthy controls or in patients with non-inflammatory neuropathy [10,11].

Classic neuropathological features of CIDP are based on neuropathological observations predominantly performed on the sural nerve. Biopsies show sural nerve infiltration of inflammatory cells CD8+ T, CD4+ T and macrophages [11]. Elevated cytokine and pro-inflammatory molecule levels in serum or cerebrospinal fluid (CSF) have also been related to CIDP, i.a., in plasma IL-12, IL-17, IL-5, IL-6, growth arrest specific 6, MMP-2 and cathepsin B with low levels of cystatin C in CSF. Compromised immune tolerance or dysfunction in regulatory leukocytes such as CD4+, CD25+, FoxP3+, and T lymphocytes have been reported, as have altered cytokine profiles towards a Th1 (and more recently Th17) phenotype. Possible B-cell maturation and polyclonal antibody synthesis, and monocyte/macrophage-mediated demyelination (direct and indirect) have been noted. The roles of T lymphocytes in the maintenance of endoneurial inflammation via cytokine secretion, as well as Schwann cell roles in potentiating the local innate and adaptive immune response with an inability to persistently terminate local inflammation by inducing T lymphocyte apoptosis have been indicated [3,10,12].

Evidence of the pathogenic role of T cells in CIDP is the increased number of circulating active T cells and elevated levels of IL-2. Another piece of evidence for the increased activity of T cells and macrophages is the increased level of TNF-α, correlating with the activity of the process. TNF may be associated with demyelination and disruption of the blood–nerve barrier [13,14]. However, the exact mechanism of nerve damage in CIDP is not fully understood. It is assumed that this mechanism is similar to that suggested in Guillain–Barré syndrome, but is much less understood. Therefore, the authors hypothesized that lymphocytes and cytokines are disturbed in patients with CIDP.

The aim of the study was to analyse the distribution of lymphocyte subpopulations and the level of cytokines in patients with the typical form of CIDP prior to starting treatment with immunoglobulins, with attention being paid to possible differences between CIDP patients with and without diabetes mellitus. The authors made an attempt to establish the correlations beteen the individual levels of lymphocytes and cytokines and electrophysiological parameters.

## 2. Materials and Methods

The study was approved by the Ethics Committee of Wroclaw Medical University in Poland. We obtained informed consent for participation in the study from all patients and volunteers.

The study group consisted of 60 patients (52 male and eight female) with a mean age of 64.8 ± 11.2 (female 59.9 ± 11.1 and male 65.5 ± 11.1), who fulfilled the diagnostic criteria for CIDP according to The European Federation of Neurological Societies (EFNS)/Peripheral Nerve Society (PNS) guidelines [15,16]. Typical phenotypic variants of CIDP were included in the study. In the group of patients with CIDP, there was a subgroup of patients with coexisting diabetes. A total of 23 patients (one female and 22 male), mean age 68.9 (female—72.0 years and male—68.7 years), had DM. The group of patients without DM consisted of 37 patients (seven female and 30 male) with a mean age of 62.3 years (female—58.1 and male—63.3 years). They were untreated with immunoglobulins and steroids. The patients were recruited from the database of our Neurological Department, and Outpatient Clinic.

The mean duration of the disease was 4.33 ± 3.04 years (minimum—1 year, maximum—19 years). In the group of patients with DM, the mean duration of the disease was 4.39 ± 3.52 years, and in the group of patients without DM—4.28 ± 2.79 years. The exclusion criteria were as follows: atypical forms of CIDP, MMN, chronic ataxic neuropathy with ophthalmoplegia, M-proteins, cold agglutinins and disialosyl antibodies (CANOMAD), POEMS syndrome (Polyneuropathy, Organomegaly, Endocrinology, Monoclonal gammopathy and Skin changes), and paraproteinaemic demyelinating neuropathy (PDN). Patients with neuropathy probably caused by B. burgdorferi infection (Lyme disease), diphtheria, or drug or toxin exposure were also excluded. According to the AAN Ad Hoc criteria, we specifically excluded patients with a cerebrospinal fluid (CSF) cell count >10/mm [16,17]. The cerebrospinal fluid for diagnostic tests was collected during a lumbar puncture. The patient was placed in a supine position, on their side, with their head bent to the chest, and lower limbs bent at the hip and knee joints and an arched spine. The injection site was in the L3/L4 intervertebral space.

### 2.1. Blood Collection and Staining for Markers of Leukocyte Subpopulations

Blood samples were taken in the group of patients and volunteers. Human peripheral blood samples were collected in preservative free EDTA tubes (BD Biosciences, San Jose, CA, USA) and immediately stained for leukocyte surface antigens with a mixture of appropriate fluorochrome-conjugated antibodies for: CD45 PerCp-Cy5.5 (clone 2D1, Sysmex Partec, Görlitz, Germany), CD3 PE-Cy7 (clone SK7, Sysmex Partec, Görlitz, Germany), CD4 FITC (clone MEM-241, Sysmex Partec, Görlitz, Germany), CD8 APC (clone MEM-31, Sysmex Partec, Görlitz, Germany), CD14 PacificBlue (clone MEM-15, Sysmex Partec, Görlitz, Germany) and CD19 PE (clone 4G7, Sysmex Partec, Görlitz, Germany). Briefly, the appropriate volume of each antibody was placed in a sterile polypropylene tube (Sarstedt) and mixed well with 50 µL of whole blood. Samples were incubated for 30–45 min at room temperature, protected from light. Then, 500 µL of VersaLyse Lysing Solution (Beckman Coulter, Immunotech, Marseille, France) was added to the samples, which were incubated for at least 15 min. Samples were acquired on a CyFlow Space flow cytometer (SysmexPartec, Görlitz, Germany) equipped with FlowMax software (Sysmex Partec, Görlitz, Germany). Six corresponding control samples (fluorescent minus one, FMO) were used to correctly set the gates of the lymphocyte subpopulations. Quantitative analysis of the leukocyte subpopulations was performed using FCS Express Flow Cytometry software (De Novo Software, Glendale, CA, USA).

### 2.2. Serum Collection, Storage and Cytokine Analysis

Human peripheral blood samples were collected in serum collection tubes (BD Biosciences, Warsaw, Poland). Blood samples were allowed to clot at room temperature for approximately 20–30 min and then centrifuged at 500× *g* for 15 min at 4 °C. The serum was immediately apportioned into 0.5 mL aliquots and transferred into clean polypropylene tubes. The serum aliquots were stored at −80 °C until cytokine analysis. A BD™ CBA Human Th1/Th2 Cytokine Kit II (BD Biosciences, San Jose, CA, USA) was used to quantitatively measure Interleukin-2 (IL-2), Interleukin-4 (IL-4), Interleukin-5 (IL-5), Interleukin-10 (IL-10), Tumor Necrosis Factor (TNF) and Interferon-γ (IFN-γ) protein levels in the serum samples. A BD™ CBA assay was performed according to the manufacturer’s procedure. The concentration of each cytokine was determined by means of a standard curve generated during the performance of the assay. The samples were acquired on CyFlow Space and CyFlow Cube flow cytometers (Sysmex-Partec, Görlitz, Germany). The results were analysed using FCAP Array v3 software (BD Biosciences, San Jose, CA, USA).

### 2.3. Electroneurography

Electrophysiological tests were performed on a Viking Quest version 10.0(Nicolet Biomedical Inc, Madison, WI, USA), Nicolet Biomedical Inc., Madison, WI, USA, device. Motor and sensory conduction tests were performed, evaluating the distal latency, amplitude and conduction velocity. In each patient, a particular nerve was examined under the same conditions and at the same distance from the stimulating cathode to the active receiving electrode and at a standardized stimulation site. Room temperature was between 21 and 23 °C, the temperature of the extremities not less than 32 °C. Compound muscle action potential (CMAP) was determined in the median, ulnar, peroneal and tibial muscles. F-wave latency, elicited by antidromic stimulation, was studied for each motor nerve. Sensory nerve action potential (SNAP) was determined in the median, ulnar and sural nerves.

## 3. Statistical Analysis

Statistical analyses (descriptive statistics, comparison of means and determination of the correlation coefficient) were performed using Statistica 13.0 software, TIBCO Software Inc., Palo Alto, CA, USA. Normality of distributions was tested using the Shapiro–Wilk test. Student’s t test was used to compare means when the variables had normal distributions and preserved homogeneity of variance, and the non-parametric Mann–Whitney U test was used for variables for which at least one subgroup did not meet the normality of distribution.

Due to the lack of normality of distribution for many variables, Spearmann’s rho rank correlation coefficient was used for analyses of relationships between variables. The level of statistical significance for all variables was set at alpha = 0.05.

In assessing correlations, the significance, sign (whether positive or negative), and strength of the relationship were evaluated. To assess the strength of the correlation the Hinkle [18] rule of thumb for interpreting the size of a correlation coefficient was used. r > 0.7 was considered a high correlation, r > 0.5 a moderate correlation, and r > 0.3 a low correlation. For the statistical analysis, we additionally divided the study group into two subgroups: those with so called “normal results” and those with “incorrect results”. An “incorrect result” was calculated as a value in the individual biochemical tests which was beyond the mean value for the whole study group ±2SD.

### Data Availability

Anonymized data not published within this article will be made available by request from any qualified investigator.

## 4. Results

In the study group, the mean level of protein in CSF was 83.31 ± 42.87. The normal range was determined as the mean + 2SD, and this was 0–169.04 mg/dL. Only five patients with typical CIDP demonstrated incorrect CSF protein levels, i.e., greater than 169.04 mg/dL.

All patients demonstrated neurophysiological features of demyelination of motor and sensory fibres with slowing of conduction velocity, conduction block in at least one nerve, dispersion of CMAPs, prolonged distal motor or sensory latencies, and prolonged F wave latencies fulfilling the EFNS/PNS criteria [16,17]. The electrodiagnostic features of demyelination with superimposed axonal degeneration are shown in Table 1.

Lymphocyte distribution and cytokine levels were determined in the blood. The percentage and total number of monocytes and lymphocytes remained normal in most of our patients. An increased percentage of monocytes was found in 14/60 patients and lymphocytes in 24/60 patients (Table 2).

Decreased levels of B CD19+ lymphocytes (10.86 ± 6.55) were observed in 41 patients; T CD3+ lymphocytes (68.53 ± 11.32) in 44 patients and T CD8+ lymphocytes (23.96 ± 10.47) in 50 patients, while in 51 patients, increased levels of CD4+ lymphocytes (65.50 ± 11.23) were seen (Figure 1).

Impaired CD4+ and CD8+ cell distribution resulted in significantly increased CD4+/CD8+ ratios (3.16 ± 2.18, range 2–14.3) in 42/60 patients. In healthy subjects, a normal range was found: monocytes 0–10% CD45+ leukocytes/200–900 cells/µL, lymphocytes 20–45% CD45+ leukocytes/1500–4500 cells/µL, B CD19+ lymphocytes 11–16% lymphocytes/60–660 cells/µL, T CD3+ lymphocytes 71–79% lymphocytes/770–2700 cells/µL, CD4+ lymphocytes 43–54% of T CD3+ lymphocytes/500–1500 cells/µL and lymphocytes CD8+ 28–37% of T CD3+ lymphocytes/270–930 cells/µL.

In CIDP patients, IL-6 (10.18 ± 4.44 vs. 8.33 ± 2.65 pg/mL, *p* = 0.019), IL-2 (7.35 ± 0.83 vs. 6.54 ± 0.85 pg/mL, *p* = 0.0006) and IL-4 (7.98 ± 0.98 vs. 7.40 ± 1.33 pg/mL, *p* = 0.047), TNF-α (10.04 ± 11.12 vs. 9.99 ± 2.92 pg/mL, *p* = 0.009) levels were significantly increased (Table 3). The levels of other cytokines, i.e., INF-ƴ (Th1 cytokine) and IL-10, were comparable in both groups.

The correlations between neurographic parameters and the level of cytokines were analysed. Serum TNF-α levels negatively correlated (low correlation) with motor nerve conduction velocity of the ulnar nerve (−0.33); IL-10 levels with CMAP amplitude of the ulnar nerve (−0.32); IL-4 levels with CMAP amplitude of the median nerve (−0.30); IL-2 levels with CMAP latency of the ulnar nerve (−0.30) and F-wave of the median nerve; and IFN-y levels (−0.30) with CMAP amplitude of the peroneal nerve. Serum IL-2 levels showed a small positive correlation with SNAP amplitude of the ulnar nerve (0.27).

The correlations between neurophysiological parameters and lymphocyte distribution showed a low, positive correlation between the total percentage of lymphocytes and CMAP amplitude of the median nerve (0.32). The numbers of B CD19+ lymphocyte cells were slightly positively correlated (little correlation) with CMAP amplitude of the median nerve (0.28). The percentage of B CD 19+ lymphocytes was positively correlated with the motor nerve conduction velocity of the ulnar nerve (0.30). The percentage of T CD3+ lymphocytes was positively correlated with CMAP amplitude of the ulnar nerve (0.28), with CMAP amplitude of the tibial nerve (0.27), motor nerve conduction velocity of the tibial nerve (0.29), and CMAP latency of the tibial nerve (0.28), but the correlations were small (little correlation). The total percentage of monocytes was negatively correlated with sensory nerve conduction velocity of the ulnar nerve (−0.32—low correlation) and sensory nerve conduction velocity of the median nerve (−0.28—little correlation).

Correlation analyses between F-wave and lymphocyte distribution were around 0.3 and showed a negative correlation between: the percentage of lymphocytes and the F-wave of the median (−0.28), the percentage of T CD3+ lymphocytes and F-wave of the tibial nerve (−0.31), and the numbers of T CD8+ lymphocyte cells with F-wave of the peroneal nerve (−0.26). Table 4 shows the correlations (r > 0.10) between individual lymphocytes, cytokines and electrophysiological parameters.

Figure 2 shows examples of three scatter plots: the percentage of CD3+ lymphocytes and CMAP amplitude of the peroneal nerve (positive correlation); serum TNF-α levels and motor conduction velocity in the ulnar nerve (negative correlation); and F-wave in the median nerve and IL-2 level (negative correlation).

Cerebrospinal fluid protein levels were significantly higher in diabetic patients (91.83 ± 46.20 vs. 78.16 ± 41.08 mg/dL; *p* = 0.049) (Figure 3). DM patients had significantly lower IL-2 levels compared to the group of patients without DM (7.01 ± 0.67 vs. 7.56 ± 0.85 pg/mL; *p* = 0.010). There were no other differences, or significant correlations between electrophysiological parameters, cytokine levels and lymphocyte distribution in patients with and without DM.

## 5. Discussion

In our study we found a markedly increased level of TNF-α, IL-6, IL-4 and IL-2. The increased level of IL-6 was accompanied with a normal production of IFN-y. Through secretion of IFN-γ and other effector molecules, Th1 cells activate macrophages, NK cells and CD8+ T cells. T cells play a pivotal role in the pathogenesis of immune-mediated nerve damage. Both CD4+ T cells and CD8+ T cells may contribute indirectly or directly to demyelination and axonal degeneration [19,20,21]. The present study found impaired CD4+ and CD8+ cell distribution in CIDP patient serum, which resulted in a significantly increased CD4+/CD8+ ratio (mean value in CIDP group was −3.16).

A higher frequency of CD4+ cells was associated with increased levels of IL-6 (Th2 cytokine) and normal levels of INF-ƴ (Th1 cytokine). IL-6 is an important pro-inflammatory cytokine produced in response to tissue injuries. Dysregulation of the permanent synthesis of IL-6 has a pathological effect in chronic inflammation and autoimmunity [22]. Mei et al. [23] showed IL-6 upregulation in CSF in CIDP patients. Increased IL-6 level probably contributes to the enhancement of local inflammation in CIDP. IL-6 may inhibit the ability of Th1 cells to secrete cytokines [24,25,26].

The theory of CIDP pathogenesis assumes a synergistic action of cell-mediated and humoral mechanisms that lead to peripheral nerve damage. The majority of our patients (44/60 patients) had incorrect results for CD3+ T lymphocytes. Other studies have also confirmed that changes to the T memory compartment are a common finding in patients with typical CIDP who were untreated with IVIg and/or steroids [27,28,29]. Sanvito et al. [30] found an increased frequency of monocytes (*p* = 0.02), decreased frequency of natural killers (*p* = 0.02), and impaired regulatory T cell function (*p* = 0.02) in CIDP compared to healthy controls. There were no significant differences in other cell populations (B cells, total and activated CD4(+) and CD8(+) T cells, effector memory and central memory CD4(+) and CD8(+) T cells, and CD4(+)CD25(high) Foxp3(+) Tregs).

Apart for the increased level of IL-6 discussed above, the levels of TNF-α, IL-4 and IL-2 were also significantly higher in our patients with CIDP. Elevated levels of inflammatory cytokines such as IL-2, IL-6, TNF-α and B-cell activating factor in serum and CSF in CIDP patients have been reported in other studies [31,32]. In CIDP, an increased number of circulating activated T cells, increased levels of IL-2 and its soluble receptors (IL-2r) are found. Pro-inflammatory cytokines (TNF-α, IFN-γ and IL-2) become expressed by a variety of cell types within the nerve and amplify the immune response [11,33]. An increased level of TNF-α is evidence of the increased activity of T cells and macrophages and, in consequence, of high activity of the inflammatory process. TNF-α may be associated with disruption of the blood–nerve barrier and demyelination [27,28]. Hagen et al. [32] demonstrated that in CIDP there is an increase in macrophage clustering around endoneurial blood vessels, as well as an increase in TNF-α staining in macrophages located in close proximity to myelinated fibres. CD4+ and particularly CD8+ T cells have been identified in inflammatory nerve infiltrates in patients with CIDP. In many studies, patients with typical CIDP show an increased level of CD8+ lymphocytes in comparison to CD4+ lymphocytes in serum and in the histopathological assessment of the sural nerve. Demyelination correlates with the presence of T lymphocytes in the nerves of CIDP patients [34,35,36].

In the literature, correlations between electrophysiological parameters and autoimmune inflammatory factors have not been demonstrated. However, we found such correlations, even though they were low or small. We found correlations between the cytokines and the percentage of lymphocytes, and electrophysiological parameters, and most interestingly, the electrophysiological parameters indicating axonal damage, i.e., CMAP and SNAP amplitudes. We found a positive correlation between the total percentage of lymphocytes, B CD 19+, CD 3+ and selective electrophysiological parameters, mainly amplitude values. Negative correlations were established between levels of TNF-α, IL-2 and electrophysiological parameters, indicating demyelinating damage as CMAP latency, conduction velocity and F wave. Negative correlations of IL-10, IL-4, and INF were found with the electrophysiological parameters, once again indicating axonal degeneration. Axonal loss in the course of CIDP has been confirmed in some studies. Mathey et al. [11], based on nerve biopsies, proved that nodal and paranodal regions are disrupted and proteins vital for maintaining structural integrity are abnormally expressed and distributed. Electron microscopic examination of nerve biopsies revealed defects in Schwann cell microvilli and paranodal glial loops with large vacuoles in the Schwann cell outer cytoplasm and nodal axoplasm. The punctate immunoreactivity for Na^+^ and K^+^ channels was distributed along the axon with diffuse distribution of caspr-1 [37,38]. The antibodies are able to penetrate into the paranodes and disrupt the CNTN1/Caspr/NF155 complex [35]. The axonal loss was associated with the inflammatory infiltrate, mainly composed of macrophages and, to a lesser extent, CD3+ T cells and CD19+ B cells with more diffuse instances of demyelination within mice nerves [9,34,39].

More severe axonal loss in nerve conduction studies and neuropathological observations has been described in patients with CIDP and diabetes mellitus (CIDP-DM). The additive effects of axonal diabetic polyneuropathy and CIDP are suspected [40]. CIDP-DM patients had higher CSF mean protein levels (91.83 vs. 78.16 mg%) and lower mean IL-2 levels (7.01 vs. 7.56 pg/mL) than CIDP patients without any other differences in the electrophysiological tests, lymphocytes distribution and cytokines levels. Zier et al. [41]. demonstrated reduced IL-2 synthesis by lymphocytes from 26 insulin-dependent diabetes mellitus (IDDM) patients compared to lymphocytes from 24 non-diabetic control subjects. These data suggest that low IL-2 synthesis is specific to IDDM and may be involved in the pathogenesis of the disease. Additionally, higher CSF protein levels than in healthy controls have also previously been observed in DM patients without CIDP [42]. Our study did not confirm unequivocal or higher pro-inflammatory risks, and worse electrophysiological findings in CIDP-DM patients.

The limited number of subjects included in the study is one of the main limitations of the analysis. The next limitation was the different CIDP duration, from 1 to 19 years, but important differences between CIDP patients and CIDP-DM patients were not seen. Nevertheless, the study allowed us to conclude that electrophysiological parameters in CIDP patients seem to be related to the autoimmune process. The analysis did not show any clear differences between patients with and without diabetes mellitus. Generally, axonal loss seemed to be secondary, but correlations we identified indicated that axonal degeneration might be independent of the demyelinating process and might be caused by direct inflammatory infiltration. The results could suggest simultaneous demyelinating and, to a lesser extent, axonal degeneration in patients with a typical form of CIDP.

## Figures and Tables

**Figure 1 jpm-11-00766-f001:**
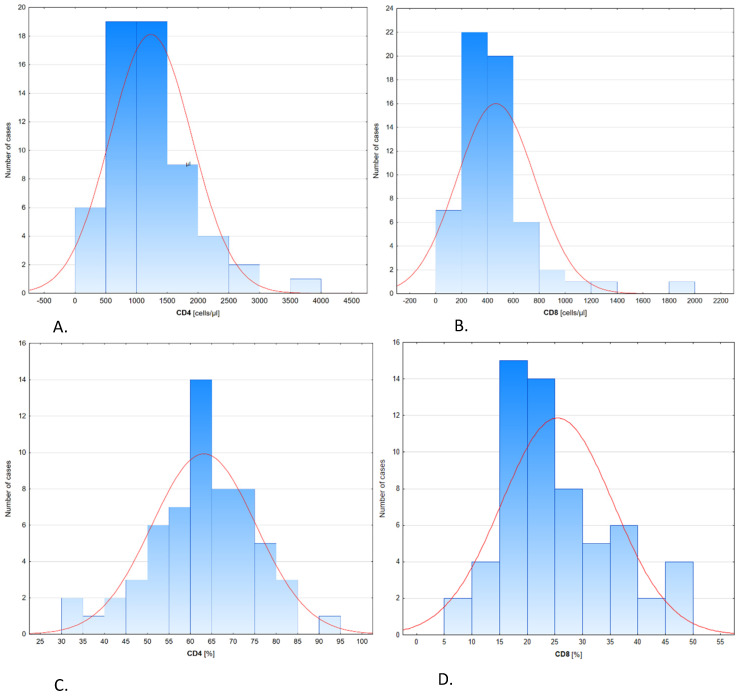
Distribution of lymphocyte subpopulations CD4+ and CD8+ in CIDP patients. Histograms of CD4+ lymphocytes (**A**), and CD8+ lymphocytes (**B**) in cells/µL, and % of CD4+ lymphocytes (**C**), and CD8+ lymphocytes (**D**).

**Figure 2 jpm-11-00766-f002:**
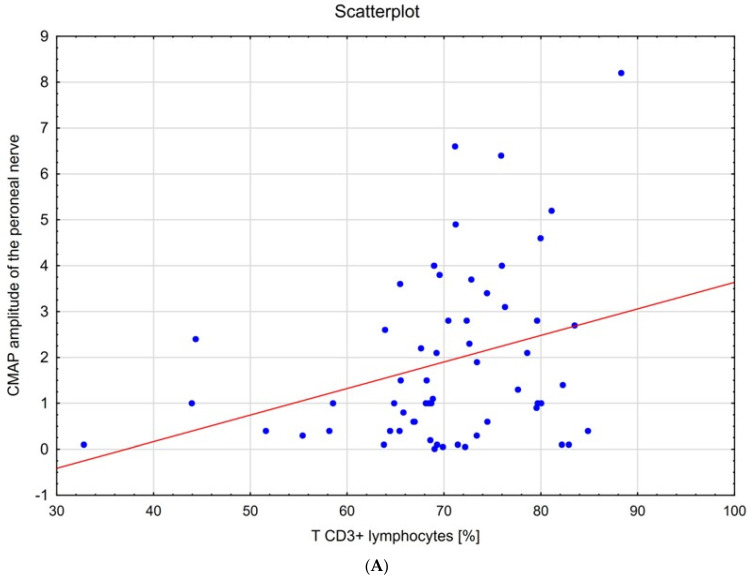
Scatter diagrams (**A**) T CD3+ lymphocytes [%]—CMAP amplitude of the peroneal nerve (**B**) serum TNF-α levels—motor conduction velocity in the ulnar nerve (**C**) F-wave in the median nerve and IL-2 level.

**Figure 3 jpm-11-00766-f003:**
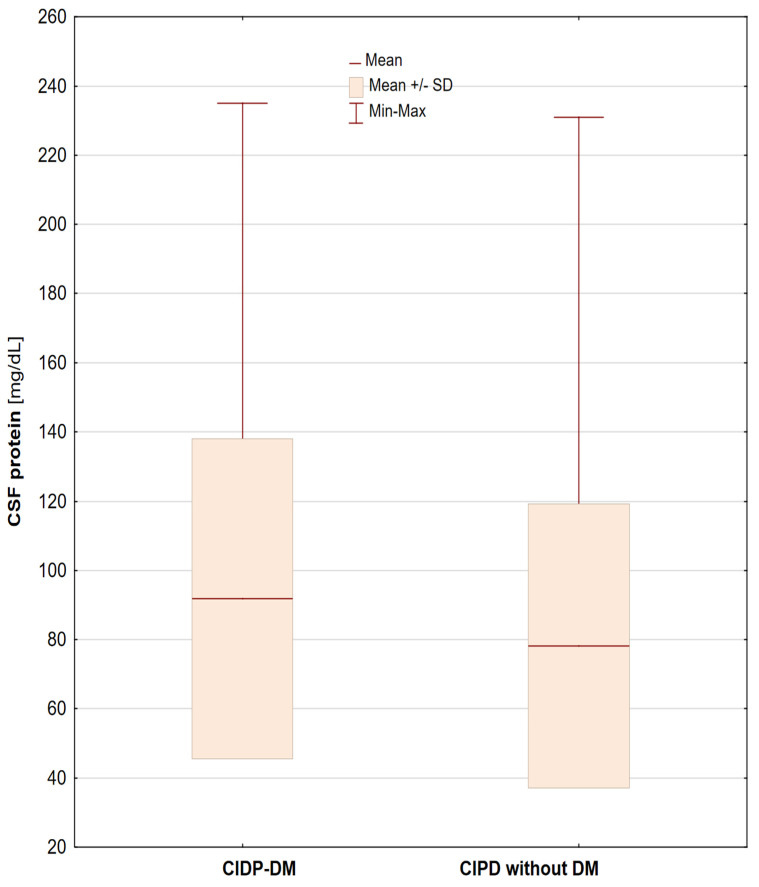
Cerebrospinal fluid protein levels in CIDP patients with diabetes mellitus (CIDP-DM) and without diabetes mellitus.

**Table 1 jpm-11-00766-t001:** Mean values of motor and sensory nerve conduction results in the study group.

		Distal Latency ± SD [ms]	Conduction Velocity ± SD [m/s]	Amplitude ± SD [mV]	F-Wave [ms]
CMAP	median	5.57 ± 3.17	46.84 ± 9.87	5.34 ± 3.42	36.24 ± 9.87
	ulnar	3.49 ± 1.48	48.07 ± 9.91	7.16 ± 2.87	35.15 ± 7.94
	peroneal	6.14 ± 1.83	34.43 ± 12.27	1.98 ± 1.95	76.33 ± 13.71
	tibial	6.37 ± 1.76	34.46 ± 11.37	2.96 ± 2.52	70.45 ± 9.29
SNAP	median	3.82 ± 0.94	36.90 ± 17.51	11.42 ± 8.97	-
	ulnar	3.54 ± 0.94	33.72 ± 17.71	11.08 ± 9.34	-
	sural	4.31 ± 1.24	25.06 ± 21.55	4.29 ± 3.33	-

CMAP: Compound muscle action potentials, SNAP: Sensory nerve action potential, SD: standard deviation, ms: millisecond, m/s: metre per second, mV: millivolt.

**Table 2 jpm-11-00766-t002:** Distribution of lymphocyte subpopulations in CIDP patients, in the whole group, and in two subgroups: with normal results (within mean value ± 2SD), and incorrect results (beyond mean value ± 2SD) of CSF proteins. SD—standard deviation, cells/µL—cells per microliter.

Leukocytes		All Patients *n* = 60	Normal Results		Incorrect Results	
		Mean ± SD	*n*	Mean ± SD	*n*	Mean ± SD
	WBC [Cells/µL]	6937.45 ± 2064.83	51	6645.29 ± 1422.33	9	8593.00 ± 3895.14
CD45+ Leukocytes	Monocytes [%]	8.01 ± 2.46	46	7.04 ± 1.87	14	11.21 ± 1.08
	Monocytes cells/µL	553.55 ± 235.37	54	513.39 ± 170.19	6	914.97 ± 415.31
	Lymphocytes [%]	39.17 ± 13.72	36	33.08 ± 6.64	24	48.30 ± 16.45
	Lymphocytes cells/µL	2702.10 ± 1208.41	45	2671.27 ± 723.27	15	2794.59 ± 2120.75
Lymphocytes	B CD19+ Lymphocytes [%]	11.54 ± 5.55	19	13.01 ± 1.54	41	10.86 ± 6.55
	B CD19+ Lymphocytes cells/µL	325.66 ± 262.29	50	290.66 ± 148.05	10	500.66 ± 539.29
	T CD3+ Lymphocytes [%]	69.98 ± 10.03	16	73.98 ± 2.31	44	68.53 ± 11.32
	T CD3+ Lymphocytes cells/µL	1894.47 ± 869.09	45	1670.97 ± 438.67	15	2564.97 ± 1391.45
T Lymphocytes	T CD3+ CD4+ Lymphocytes [%]	62.97 ± 12.05	9	48.69 ± 3.58	51	65.50 ± 11.23
	T CD3+ CD4+ Lymphocytes cells/µL	1223.41 ± 660.56	38	1026.65 ± 295.22	22	1563.28 ± 938.76
	T CD3+ CD8+ Lymphocytes [%]	25.33 ± 10.09	10	32.19 ± 2.78	50	23.96 ± 10.47
	T CD3+ CD8+ Lymphocytes cells/µL	461.82 ± 299.29	42	468.15 ± 153.12	18	447.06 ± 488.45
	CD4/CD8	3.16 ± 2.18	6	2.02 ± 0.12	54	3.28 ± 2.27

**Table 3 jpm-11-00766-t003:** Mean values and standard deviations of cytokines in the patient and control groups.

Cytokines	CIDP Patients *n* = 60	Control Group *n* = 18	*p*-Value
	Mean ± SD [pg/mL]	Mean ± SD [pg/mL]	
IFN-y	7.72 ± 1.31	7.29 ± 1.96	0.590
TNF-α	10.04 ± 11.12	9.99 ± 2.92	0.009
IL-10	8.15 ± 1.12	7.77 ± 1.75	0.524
IL-6	10.18 ± 4.44	8.33 ± 2.65	0.019
IL-4	7.98 ± 0.98	7.40 ± 1.33	0.047
IL-2	7.35 ± 0.83	6.54 ± 0.85	0.0006

**Table 4 jpm-11-00766-t004:** Correlations (r > 0.10) between individual lymphocytes, cytokines and electrophysiological parameters (CMAP and SNAP).

			CMAP Amplitude				SNAP Amplitude			F-Wave			
			Median	Ulnar	Peroneal	Tibial	Median	Ulnar	Sural	Median	Ulnar	Peroneal	Tibial
Lymphocytes	Monocytes	[%]							−0.17				
		cells/µL				−0.11	−0.23		−0.19		0.12		
	Lymphocytes	[%]	**0.32**						−0.18	−0.28		−0.14	−0.12
		cells/µL					−0.13	−0.17	−0.19	−0.14			
	B CD19+ Lymphocytes	[%]	0.24	−0.14	−0.24	−0.19	0.14	0.14				0.13	
		cells/µL	0.26	−0.14	−0.21	−0.12							
	T CD3+ Lymphocytes	[%]	0.16	0.22	0.28	0.27	0.11			−0.20	−0.17	−0.20	**−0.31**
		cells/µL						−0.15	−0.22	−0.17		−0.13	−0.16
	TCD3+ CD4+ Lymphocytes	[%]	0.16							−0.14	−0.14		−0.14
		cells/µL	0.11				−0.11		−0.16	−0.19		−0.11	−0.17
	TCD3+ CD8+ Lymphocytes	[%]											0.12
		cells/µL	0.13			0.14				−0.15		−0.26	
Cytokines	IFN-y		−0.24	−0.19	**−0.30**			0.25	−0.13	−0.24			
	TNF-α		−0.20	−0.25	−0.16					−0.18			
	IL-10		−0.13	**−0.32**	−0.28	0.13			−0.12				
	IL-6		−0.25	−0.17	−0.24		−0.15		−0.13	−0.20			
	IL-4		**−0.30**		−0.21			0.19		−0.24			
	IL-2		−0.24				0.14	0.27		**−0.37**	−0.22	−0.12	

CMAP—Compound muscle action potentials, SNAP—Sensory nerve action potential, cells/µL—cells per microliter, SD—standard deviation.

## Data Availability

No new data were created or analyzed in this study. Data sharing is not applicable to this article.

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
