# Peer review of "Correlations between Electrophysiological Parameters, Lymphocyte Distribution and Cytokine Levels in Patients with Chronic Demyelinating Inflammatory Polyneuropathy"

_jpm, 2021, doi:10.3390/jpm11080766_

Round 1

Reviewer 1 Report

The authors found a weak correlation between CMAP amplitudes and cytokine levels. The study is interesting and reinforces the hypothesis of immune participation in the disease, although due to the weakness of the relationship it is suspected that environmental factors must also participate, a circumstance that should be further explained in the text.

The title should not include abbreviations that have not been explained before. Please change it to “Correlations between electrophysiological parameters, lymphocyte distribution and cytokine levels in patients with chronic demyelinating inflammatory polineuropathy”. If it is too long, simplify other words but not the name of the disease. Introduction: It is not clear what the reasons were for a sub-study with type 2 diabetes.

Methods: The reasons for creating a diabetic subgroup are also not explained. In a new version of the manuscript, in the method section when talking about the Study Population, both the reasons for creating this subgroup and the characteristics of the subgroups should be explained in a way that is easily understood by readers.

Results: Although in the statistical methods section the authors describe that they use the Guilford's rule of thumb to evaluate the strength of the correlation, the results do not use this classification. Please rate the degree of correlation and use this rule where appropriate.

Protein levels would be better understood if they were plotted on a graph. It can be made of violin or boxes and whiskers representing the outliers.

When the authors describe the percentage and the total number of monocytes and lymphocytes, although it is assumed that these values ​​refer to perioral blood, it is not expressed. Please clarify in the manuscript where the studied sample came from (blood or CSF) so that there are no doubts among the readers.

The section where the authors discuss the altered distribution of CD4 + and CD8 cells is confusingly described. There is talk of a significant increase but the level of significance (p value) is not indicated. Likewise, a range 2-14.3 is described without its meaning being clear. A graphic representation made it easier to understand what the authors want to explain.

Results of cytokine quantifications: Although they are described in the tables, in the most important aspects of the work, such as this one, the differences and the significance value (p value) must be reflected in the text. In this sense, the IL6 and TNF-alpha values ​​should also be reflected in the main text of the article. The results of Table 3 should be represented in a box-and-whisker plot to have more information on the dispersion of the values.

When authors describe the correlation, it is not enough to write positive or negative correlation, and if it is significant. The degree of correlation must also be reported using a rule of thumb for interpreting the size of a correlation coefficient. As the authors describe in methods that use Gilford's rule, here they should say that it is “low correlation”. If this qualification seems to the authors that it may detract from their results, I suggest that they use another rule such as those described by Hinkle (2003), Romsey (2011) or Mukaka (2012), indicating the degree of correlation and its significance (p value).

In the description of the association of CMAP and the percentage of lymphocytes, the reviewer understands that they are still referring to peripheral blood levels, but a reader of the work may not be so clear. Please clearly state that it is blood so that readers have no doubts. The discussion of IL-2 levels in patients with DM should better reflect the value of the findings in clinical practice. In the text it is mentioned that patients with DM have IL-2 levels "significantly" lower than patients without DM (7.01 vs 7.56). These differences are discrete and are within the ranges of normality. Please indicate this fact to avoid possible confusion for readers. The fact that there are statistically significant differences does not mean that they are clinically relevant, in the same way that if there is a weight difference of 100 grams between two groups of people (65kg vs 65.1 kg) and that this difference is significant, it should be avoided that any reader can get to interpret that the patients of the second group are “obese”.

Discussion :

It should start with a summary of the main findings of the authors and their possible relevance to the scientific community. A continuation of these findings is when a comparison should be made with the findings of other authors in the literature and with the various biological models. The phrase: "In our study, the increased ability of Th2 cells to produce IL-6 was accompanied…." must be rewritten. The study does not show which cells secrete IL6 and therefore it cannot be said that TH2 have a greater capacity to secrete IL6. Although IL6 is a cytokine of the adaptive immune response TH2, it is also a cytokine that regulates the processes of innate immunity in the early stages of acute infection and during chronic processes. As the authors  comment later, IL6 is produced by TH2 and also other cells as monocytes, macrophages, fibroblasts and endothelial cells. In order to demonstrate a greater secretory capacity of IL6 in the T lymphocytes of patients, intracellular IL6 should have been evaluated and compared with controls, or functional “in vitro” studies should be carried out. As this cannot be done, the reference to increased IL6 secretion by TH2 should be removed from the text.

In the conclusions it is stated that: “… electrophysiological parameters in CIDP patients are closely related to the autoimmune process”. This statement is too categorical and should be softened to fit more closely with reality. With the results obtained (correlation coefficients between 0.26 and 0.32), a term as strong as “closely” cannot be used. If the degree of the association is described, the qualification of Gilford's rule of thumbs should be used, (which is the one mentioned in methods) and qualify as: “small relationship”.

It is recommended that the new version of the manuscript be reviewed by an immunologist and statistician before being sent.

Author Response

Comments and Suggestions for Authors

The authors found a weak correlation between CMAP amplitudes and cytokine levels. The study is interesting and reinforces the hypothesis of immune participation in the disease, although due to the weakness of the relationship it is suspected that environmental factors must also participate, a circumstance that should be further explained in the text. – added to the text – page 3.

The title should not include abbreviations that have not been explained before. Please change it to “Correlations between electrophysiological parameters, lymphocyte distribution and cytokine levels in patients with chronic demyelinating inflammatory polineuropathy”. title changed - disease abbreviation changed to full name If it is too long, simplify other words but not the name of the disease.

Introduction: It is not clear what the reasons were for a sub-study with type 2 diabetes. the mutual correlation / comorbidity of diabetes and CIDP was explained, information added to the introduction

Methods: The reasons for creating a diabetic subgroup are also not explained. In a new version of the manuscript, in the method section when talking about the Study Population, both the reasons for creating this subgroup and the characteristics of the subgroups should be explained in a way that is easily understood by readers. The Method explains the additional division of the CIDP group into diabetic and non-diabetic patients.

Results: Although in the statistical methods section the authors describe that they use the Guilford's rule of thumb to evaluate the strength of the correlation, the results do not use this classification. Please rate the degree of correlation and use this rule where appropriate. – It was changed to Hinkle interpretation page 7, and later used in the Result section.

Protein levels would be better understood if they were plotted on a graph. It can be made of violin or boxes and whiskers representing the outliers. We added Fig. 3 presenting the protein level in CIDP-DM and CIDP patients without DM.

When the authors describe the percentage and the total number of monocytes and lymphocytes, although it is assumed that these values ​​refer to perioral blood, it is not expressed. Please clarify in the manuscript where the studied sample came from (blood or CSF) so that there are no doubts among the readers. - The method explained that the distribution of lymphocytes and the level of cytokines were determined in the blood sample.

The section where the authors discuss the altered distribution of CD4 + and CD8 cells is confusingly described. There is talk of a significant increase but the level of significance (p value) is not indicated.- added on the page 10. Likewise, a range 2-14.3 is described without its meaning being clear. – value was added – page 8.  and A graphic representation made it easier to understand what the authors want to explain. – Graphic presentation of CD4+ and CD8+ were added (Fig.1).

Results of cytokine quantifications: Although they are described in the tables, in the most important aspects of the work, such as this one, the differences and the significance value (p value) must be reflected in the text. In this sense, the IL6 and TNF-alpha values ​​should also be reflected in the main text of the article. Added: values for CIDP versus control group, and p value – 8 page  The results of Table 3 should be represented in a box-and-whisker plot to have more information on the dispersion of the values.

When authors describe the correlation, it is not enough to write positive or negative correlation, and if it is significant. The degree of correlation must also be reported using a rule of thumb for interpreting the size of a correlation coefficient. As the authors describe in methods that use Gilford's rule, here they should say that it is “low correlation”. If this qualification seems to the authors that it may detract from their results, I suggest that they use another rule such as those described by Hinkle (2003), Romsey (2011) or Mukaka (2012), indicating the degree of correlation and its significance (p value). As above: it was changed to Hinkle interpretation page 7, and later used in the Result section with descriptions.

In the description of the association of CMAP and the percentage of lymphocytes, the reviewer understands that they are still referring to peripheral blood levels, but a reader of the work may not be so clear. Please clearly state that it is blood so that readers have no doubts. The discussion of IL-2 levels in patients with DM should better reflect the value of the findings in clinical practice. In the text it is mentioned that patients with DM have IL-2 levels "significantly" lower than patients without DM (7.01 vs 7.56). These differences are discrete and are within the ranges of normality. Please indicate this fact to avoid possible confusion for readers. The fact that there are statistically significant differences does not mean that they are clinically relevant, in the same way that if there is a weight difference of 100 grams between two groups of people (65kg vs 65.1 kg) and that this difference is significant, it should be avoided that any reader can get to interpret that the patients of the second group are “obese”. The problem with the slight differences between CIDP patients and CIDP-DM patients, and the possible meaning of the differences was explain in the text (page 11)

Discussion :

It should start with a summary of the main findings of the authors and their possible relevance to the scientific community. A continuation of these findings is when a comparison should be made with the findings of other authors in the literature and with the various biological models. The phrase: "In our study, the increased ability of Th2 cells to produce IL-6 was accompanied…." must be rewritten. The study does not show which cells secrete IL6 and therefore it cannot be said that TH2 have a greater capacity to secrete IL6. Although IL6 is a cytokine of the adaptive immune response TH2, it is also a cytokine that regulates the processes of innate immunity in the early stages of acute infection and during chronic processes. As the authors  comment later, IL6 is produced by TH2 and also other cells as monocytes, macrophages, fibroblasts and endothelial cells. In order to demonstrate a greater secretory capacity of IL6 in the T lymphocytes of patients, intracellular IL6 should have been evaluated and compared with controls, or functional “in vitro” studies should be carried out. As this cannot be done, the reference to increased IL6 secretion by TH2 should be removed from the text.

In the conclusions it is stated that: “… electrophysiological parameters in CIDP patients are closely related to the autoimmune process”. This statement is too categorical and should be softened to fit more closely with reality. With the results obtained (correlation coefficients between 0.26 and 0.32), a term as strong as “closely” cannot be used. If the degree of the association is described, the qualification of Gilford's rule of thumbs should be used, (which is the one mentioned in methods) and qualify as: “small relationship”

The Discussion part was partially rewritten, and rearranged,  as suggested by the reviewer 1 and 2.

It is recommended that the new version of the manuscript be reviewed by an immunologist and statistician before being sent. ??????? The manuscript was reviewed by immunologist, statistician, and English native speaker.

Submission Date

30 June 2021

Date of this review

12 Jul 2021 13:04:26

Reviewer 2 Report

In this article, the aim of the authors was to investigate the correlations existing between electrophysiological parameters, lymphocyte distribution and cytokine levels in patients with chronic inflammatory demyelinating polyneuropathy (CIDP). The disease is complex and can take various forms; CIDP is therefore rather considered as a syndrome. There may be a number of different causes of the disorder, which manifests in different ways. Any attempts to better understand the multiple aetiology of CIDP is therefore important for elaborate better treatments and follow-up of this rare disease.

Below are some specific concerns regarding this study and the manuscript.

  1. The introduction of this manuscript is relatively short, essentially centred on a clinical description of the disease and poorly supported scientifically by previous references. Most importantly, the goals and objectives of the author’s study are not clearly highlighted (one single sentence without scientific question). This section should be significantly developed, with much detailed scientific support and presented in a more logical way. It should be more focused on the scientific questions that are asked.
  2. There are abbreviations that are not described (two in the abstract, CIDP and CMAP) or too late like IL (in pages 4 and again in page 11), some others are described several times, e.g. MMM in page 2 and 3;
  3. M&M: how the CSF was collected is not described (used in page 5)
  4. M& M, subsection 2.1: the isotype controls are not described. Is it what is described as FMO? The references of each reagent should be indicated.
  5. Results: it is unclear if the data that are described were obtained in the blood or in the CSF (pages 6 and 7). The tables have no legend (just a title).
  6. The data with correlation values should be shown in a series of figures
  7. Why DM patients were used is not justified. What was the indication for collecting CSF from these patients (page 9). These patients were not described in page 3. How many were included, what were their characteristics?
  8. There is no legend to the Fig. 1. This figure should be described in much details in the text.
  9. Discussion: this section is highly speculative, purely correlative without any mechanistic support that may reinforce the hypotheses. Association doesn’t mean cause and effect. This section is long, not focused and finally brings no novel, scientifically strong information.

v

Author Response

In this article, the aim of the authors was to investigate the correlations existing between electrophysiological parameters, lymphocyte distribution and cytokine levels in patients with chronic inflammatory demyelinating polyneuropathy (CIDP). The disease is complex and can take various forms; CIDP is therefore rather considered as a syndrome. There may be a number of different causes of the disorder, which manifests in different ways. Any attempts to better understand the multiple aetiology of CIDP is therefore important for elaborate better treatments and follow-up of this rare disease.

Below are some specific concerns regarding this study and the manuscript.

  1. The introduction of this manuscript is relatively short, essentially centred on a clinical description of the disease and poorly supported scientifically by previous references. Most importantly, the goals and objectives of the author’s study are not clearly highlighted (one single sentence without scientific question). This section should be significantly developed, with much detailed scientific support and presented in a more logical way. It should be more focused on the scientific questions that are asked. – the introduction was changed, was put in order, with attention to diabetes mellitus in CIDP patients.
  2. There are abbreviations that are not described (two in the abstract, CIDP and CMAP) or too late like IL (in pages 4 and again in page 11), some others are described several times, e.g. MMM in page 2 and 3; - as indicated in the tip, the abbreviations and full names were corrected
  3. M&M: how the CSF was collected is not described (used in page 5) - a method of performing a lumbar puncture was added
  4. M& M, subsection 2.1: the isotype controls are not described. Is it what is described as FMO? The references of each reagent should be indicated.- the explanation is below
  5. Results: it is unclear if the data that are described were obtained in the blood or in the CSF (pages 6 and 7). The tables have no legend (just a title). in M&M and Results it was emphasized that the distribution of lymphocytes and the level of cytokines were determined in the blood sample.
  6. The data with correlation values should be shown in a series of figures
  7. Why DM patients were used is not justified. What was the indication for collecting CSF from these patients (page 9). These patients were not described in page 3. How many were included, what were their characteristics? - In M&M, two subgroups of patients with CIDP were clearly highlighted - diabetic and non-diabetic
  8. There is no legend to the Fig. 1. This figure should be described in much details in the text. – details were put into the text - page 9.
  9. Discussion: this section is highly speculative, purely correlative without any mechanistic support that may reinforce the hypotheses. Association doesn’t mean cause and effect. This section is long, not focused and finally brings no novel, scientifically strong information. –

The Discussion part was partially rewritten, and rearranged,  as suggested by the reviewer 1 and 2.

 Ad. 4

To properly identify and gate subpopoulation of leucocytes and limfocytes, we used FMO Fluorescence Minus One Control controls, which are the most useful and adequate for multicolor analysis using flow cytometry. An FMO controls contains the same antibodies (conjugated with flurochromes) as are used  in a panel for blood sample analysis,  except for the one that is being measured.  It ensure to set good gates for cell subpopulating being analysed. The use of isotope controls has many limitations and thus its use is very controversial(https://www.bio-rad-antibodies.com/flow-cytometry-isotype-controls.html;https://expert.cheekyscientist.com/when-to-use-and-not-use-flow-cytometry-isotype-controls/; Holden T. Maecker, Joseph Trotter ”Flow cytometry controls, instrument setup, and the determination of positivity” doi.org/10.1002/cyto.a.20333). The isotype control, to be useful to identify nonspecific biding , should be the same isotype ( in terms of species, heavy chain and light chain), the same fluorochrome and have the same fluorochrom:protein ratio. Unfortunately, there is no such perfect isotype control on the market. For these reasons, experts in the field of flow cytometry are moving beyond the isotype control.

Below is the gating strategy used in cytometric analysis

Round 2

Reviewer 1 Report

In the answers the authors say that they have placed the p-value on page 10. The new manuscript does not have the pages numbered. If we sequentially assign numbers from the first page, the 10th page contains Figure 1 and does not contain any p-values. Possibly it is an error when indicating the location of the change since the references to the changes on page 8 correspond to what is described on the page that is in eighth place of the new version of the manuscript.

Author Response

Reviewer 1:

All changes are in yellow in the manuscript.

PoczÄ…tek formularza

Comments and Suggestions for Authors

The authors found a weak correlation between CMAP amplitudes and cytokine levels. The study is interesting and reinforces the hypothesis of immune participation in the disease, although due to the weakness of the relationship it is suspected that environmental factors must also participate, a circumstance that should be further explained in the text. – added to the text – page 3.

The title should not include abbreviations that have not been explained before. Please change it to “Correlations between electrophysiological parameters, lymphocyte distribution and cytokine levels in patients with chronic demyelinating inflammatory polineuropathy”. title changed - disease abbreviation changed to full name If it is too long, simplify other words but not the name of the disease.

Introduction: It is not clear what the reasons were for a sub-study with type 2 diabetes. the mutual correlation / comorbidity of diabetes and CIDP was explained, information added to the introduction

Methods: The reasons for creating a diabetic subgroup are also not explained. In a new version of the manuscript, in the method section when talking about the Study Population, both the reasons for creating this subgroup and the characteristics of the subgroups should be explained in a way that is easily understood by readers. The Method explains the additional division of the CIDP group into diabetic and non-diabetic patients.

Results: Although in the statistical methods section the authors describe that they use the Guilford's rule of thumb to evaluate the strength of the correlation, the results do not use this classification. Please rate the degree of correlation and use this rule where appropriate. – It was changed to Hinkle interpretation page 7, and later used in the Result section.

Protein levels would be better understood if they were plotted on a graph. It can be made of violin or boxes and whiskers representing the outliers. We added Fig. 3 presenting the protein level in CIDP-DM and CIDP patients without DM.

When the authors describe the percentage and the total number of monocytes and lymphocytes, although it is assumed that these values ​​refer to perioral blood, it is not expressed. Please clarify in the manuscript where the studied sample came from (blood or CSF) so that there are no doubts among the readers. - The method explained that the distribution of lymphocytes and the level of cytokines were determined in the blood sample.

The section where the authors discuss the altered distribution of CD4 + and CD8 cells is confusingly described. There is talk of a significant increase but the level of significance (p value) is not indicated.- added on the page 10. Likewise, a range 2-14.3 is described without its meaning being clear. – value was added – page 8.  and A graphic representation made it easier to understand what the authors want to explain. – Graphic presentation of CD4+ and CD8+ were added (Fig.1).

Results of cytokine quantifications: Although they are described in the tables, in the most important aspects of the work, such as this one, the differences and the significance value (p value) must be reflected in the text. In this sense, the IL6 and TNF-alpha values ​​should also be reflected in the main text of the article. Added: values for CIDP versus control group, and p value – 8 page  The results of Table 3 should be represented in a box-and-whisker plot to have more information on the dispersion of the values.

When authors describe the correlation, it is not enough to write positive or negative correlation, and if it is significant. The degree of correlation must also be reported using a rule of thumb for interpreting the size of a correlation coefficient. As the authors describe in methods that use Gilford's rule, here they should say that it is “low correlation”. If this qualification seems to the authors that it may detract from their results, I suggest that they use another rule such as those described by Hinkle (2003), Romsey (2011) or Mukaka (2012), indicating the degree of correlation and its significance (p value). As above: it was changed to Hinkle interpretation page 7, and later used in the Result section with descriptions.

In the description of the association of CMAP and the percentage of lymphocytes, the reviewer understands that they are still referring to peripheral blood levels, but a reader of the work may not be so clear. Please clearly state that it is blood so that readers have no doubts. The discussion of IL-2 levels in patients with DM should better reflect the value of the findings in clinical practice. In the text it is mentioned that patients with DM have IL-2 levels "significantly" lower than patients without DM (7.01 vs 7.56). These differences are discrete and are within the ranges of normality. Please indicate this fact to avoid possible confusion for readers. The fact that there are statistically significant differences does not mean that they are clinically relevant, in the same way that if there is a weight difference of 100 grams between two groups of people (65kg vs 65.1 kg) and that this difference is significant, it should be avoided that any reader can get to interpret that the patients of the second group are “obese”. The problem with the slight differences between CIDP patients and CIDP-DM patients, and the possible meaning of the differences was explain in the text (page 11)

Discussion :

It should start with a summary of the main findings of the authors and their possible relevance to the scientific community. A continuation of these findings is when a comparison should be made with the findings of other authors in the literature and with the various biological models. The phrase: "In our study, the increased ability of Th2 cells to produce IL-6 was accompanied…." must be rewritten. The study does not show which cells secrete IL6 and therefore it cannot be said that TH2 have a greater capacity to secrete IL6. Although IL6 is a cytokine of the adaptive immune response TH2, it is also a cytokine that regulates the processes of innate immunity in the early stages of acute infection and during chronic processes. As the authors  comment later, IL6 is produced by TH2 and also other cells as monocytes, macrophages, fibroblasts and endothelial cells. In order to demonstrate a greater secretory capacity of IL6 in the T lymphocytes of patients, intracellular IL6 should have been evaluated and compared with controls, or functional “in vitro” studies should be carried out. As this cannot be done, the reference to increased IL6 secretion by TH2 should be removed from the text.

In the conclusions it is stated that: “… electrophysiological parameters in CIDP patients are closely related to the autoimmune process”. This statement is too categorical and should be softened to fit more closely with reality. With the results obtained (correlation coefficients between 0.26 and 0.32), a term as strong as “closely” cannot be used. If the degree of the association is described, the qualification of Gilford's rule of thumbs should be used, (which is the one mentioned in methods) and qualify as: “small relationship”

The Discussion part was partially rewritten, and rearranged,  as suggested by the reviewer 1 and 2.

It is recommended that the new version of the manuscript be reviewed by an immunologist and statistician before being sent. ??????? The manuscript was reviewed by immunologist, statistician, and English native speaker.

Submission Date

30 June 2021

Date of this review

12 Jul 2021 13:04:26

DóÅ‚ formularza

© 1996-2021 MDPI (Basel, Switzerland) unless otherwise stated

Reviewer 2 Report

I have no comment to the Authors

Author Response

Reviewer 2

All changes are in yellow in the manuscript.

In this article, the aim of the authors was to investigate the correlations existing between electrophysiological parameters, lymphocyte distribution and cytokine levels in patients with chronic inflammatory demyelinating polyneuropathy (CIDP). The disease is complex and can take various forms; CIDP is therefore rather considered as a syndrome. There may be a number of different causes of the disorder, which manifests in different ways. Any attempts to better understand the multiple aetiology of CIDP is therefore important for elaborate better treatments and follow-up of this rare disease.

Below are some specific concerns regarding this study and the manuscript.

  1. The introduction of this manuscript is relatively short, essentially centred on a clinical description of the disease and poorly supported scientifically by previous references. Most importantly, the goals and objectives of the author’s study are not clearly highlighted (one single sentence without scientific question). This section should be significantly developed, with much detailed scientific support and presented in a more logical way. It should be more focused on the scientific questions that are asked. – the introduction was changed, was put in order, with attention to diabetes mellitus in CIDP patients.
  2. There are abbreviations that are not described (two in the abstract, CIDP and CMAP) or too late like IL (in pages 4 and again in page 11), some others are described several times, e.g. MMM in page 2 and 3; - as indicated in the tip, the abbreviations and full names were corrected
  3. M&M: how the CSF was collected is not described (used in page 5) - a method of performing a lumbar puncture was added
  4. M& M, subsection 2.1: the isotype controls are not described. Is it what is described as FMO? The references of each reagent should be indicated.- the explanation is below
  5. Results: it is unclear if the data that are described were obtained in the blood or in the CSF (pages 6 and 7). The tables have no legend (just a title). in M&M and Results it was emphasized that the distribution of lymphocytes and the level of cytokines were determined in the blood sample.
  6. The data with correlation values should be shown in a series of figures
  7. Why DM patients were used is not justified. What was the indication for collecting CSF from these patients (page 9). These patients were not described in page 3. How many were included, what were their characteristics? - In M&M, two subgroups of patients with CIDP were clearly highlighted - diabetic and non-diabetic
  8. There is no legend to the Fig. 1. This figure should be described in much details in the text. – details were put into the text - page 9.
  9. Discussion: this section is highly speculative, purely correlative without any mechanistic support that may reinforce the hypotheses. Association doesn’t mean cause and effect. This section is long, not focused and finally brings no novel, scientifically strong information. –

The Discussion part was partially rewritten, and rearranged,  as suggested by the reviewer 1 and 2.

 Ad. 4

To properly identify and gate subpopoulation of leucocytes and limfocytes, we used FMO Fluorescence Minus One Control controls, which are the most useful and adequate for multicolor analysis using flow cytometry. An FMO controls contains the same antibodies (conjugated with flurochromes) as are used  in a panel for blood sample analysis,  except for the one that is being measured.  It ensure to set good gates for cell subpopulating being analysed. The use of isotope controls has many limitations and thus its use is very controversial(https://www.bio-rad-antibodies.com/flow-cytometry-isotype-controls.html;https://expert.cheekyscientist.com/when-to-use-and-not-use-flow-cytometry-isotype-controls/; Holden T. Maecker, Joseph Trotter ”Flow cytometry controls, instrument setup, and the determination of positivity” doi.org/10.1002/cyto.a.20333). The isotype control, to be useful to identify nonspecific biding , should be the same isotype ( in terms of species, heavy chain and light chain), the same fluorochrome and have the same fluorochrom:protein ratio. Unfortunately, there is no such perfect isotype control on the market. For these reasons, experts in the field of flow cytometry are moving beyond the isotype control.

Below is the gating strategy used in cytometric analysis
